# Effect of a 90 g/day low-carbohydrate diet on glycaemic control, small, dense low-density lipoprotein and carotid intima-media thickness in type 2 diabetic patients: An 18-month randomised controlled trial

Chin-Ying Chen[1], Wei-Sheng Huang[1]*, Hui-Chuen Chen[2], Chin-Hao Chang[3], Long-Teng Lee[1], Heng-Shuen Chen[1], Yow-Der Kang[1], Wei-Chu Chie[4], Chyi-Feng Jan[1], Wei-Dean Wang[1], Jaw-Shiun Tsai[1]

1 Department of Family Medicine, National Taiwan University Hospital & National Taiwan University, Taipei, Taiwan, 2 Department of Dietetics, National Taiwan University Hospital & National Taiwan University, Taipei, Taiwan, 3 Department of Medical Research, National Taiwan University Hospital & National Taiwan University, Taipei, Taiwan, 4 Department of Public Health, College of Public Health, Institute of Epidemiology and Preventive Medicine, National Taiwan University, Taipei, Taiwan

* starsambora@gmail.com

## Abstract

### Aim

This study explored the effect of a moderate (90 g/d) low-carbohydrate diet (LCD) in type 2 diabetes patients over 18 months.

### Methods

Ninety-two poorly controlled type 2 diabetes patients aged 20–80 years with HbA1c ≥7.5% (58 mmol/mol) in the previous three months were randomly assigned to a 90 g/d LCD r traditional diabetic diet (TDD). The primary outcomes were glycaemic control status and change in medication effect score (MES). The secondary outcomes were lipid profiles, small, dense low-density lipoprotein (sdLDL), serum creatinine, microalbuminuria and carotid intima-media thickness (IMT).

### Results

A total of 85 (92.4%) patients completed 18 months of the trial. At the end of the study, the LCD and TDD group consumed 88.0±29.9 g and 151.1±29.8 g of carbohydrates, respectively (p < 0.05). The 18-month mean change from baseline was statistically significant for the HbA1c (-1.6±0.3 vs. -1.0±0.3%), 2-h glucose (-94.4±20.8 vs. -18.7±25.7 mg/dl), MES (-0.42±0.32 vs. -0.05±0.24), weight (-2.8±1.8 vs. -0.7±0.7 kg), waist circumference (-5.7 ±2.7 vs. -1.9±1.4 cm), hip circumference (-6.1±1.8 vs. -2.9±1.7 cm) and blood pressure (-8.3±4.6/-5.0±3 vs. 1.6±0.5/2.5±1.6 mmHg) between the LCD and TDD groups (p<0.05). The 18-month mean change from baseline was not significantly different in lipid profiles,

**Data Availability Statement:** All relevant data are within the manuscript and its Supporting Information files.

**Funding:** The project was funded by the National Taiwan University Hospital project number 106-37. The funder had no role in study design, data collection and analysis, decision to publish, or preparation of the manuscript.

**Competing interests:** The authors have declared that no competing interests exist.

sdLDL, serum creatinine, microalbuminuria, alanine aminotransferase (ALT) and carotid IMT between the groups.

## Conclusions

A moderate (90 g/d) LCD showed better glycaemic control with decreasing MES, lowering blood pressure, decreasing weight, waist and hip circumference without adverse effects on lipid profiles, sdLDL, serum creatinine, microalbuminuria, ALT and carotid IMT than TDD for type 2 diabetic patients.

## Introduction

Most diabetic organisations recommend a traditional diabetic diet (TDD) with a carbohydrate intake of 50–60%, protein intake of 1.0–1.2 g/kg and a total fat intake of $\leq$ 30% of total energy. However, there is insufficient evidence to justify these recommendations [1–3].

There is increasing evidence of the effectiveness of a low-carbohydrate diet (LCD) (carbohydrate < 130 g/day or < 26% of total energy intake) for weight loss and glycaemic control of type 2 diabetes mellitus (DM) [4–6]. LCDs provide short-term (three to six month) improvements in glycaemic control, weight loss and lower cardiovascular risk for people with type 2 DM; however, this improvement is not sustained over longer periods (one to two years) [7, 8]. The difference may be because of the inconsistent definitions of LCD [9], the underestimated effect of decreasing medication [10–12] and achievement of dietary goals [10, 13]. Most LCD studies have involved a VLCD (very low-carbohydrate diet) for six months or less [5, 11, 14], save for two 12-month studies [10, 12]. Using a moderate LCD, one recent 130 g/day LCD study [13] showed effectiveness at six months, but two earlier studies showed no significant effect for such LCD in at the one-year or two-year follow-up [15, 16].

A VLCD was effective in the short term but difficult when it came to achieving long-term goals [9, 10]. Although there is no standard definition of a low-carbohydrate diet, a VLCD involves 20–50 g/day or < 10% of the 2000 kcal/day diet [7]. In VLCD trials, people often deviate from the carbohydrate goal, consuming an average of 19–25 g more than the recommended quantity in the short term [10, 14] and up to 132–162 g in the long term [9]. A practical moderate LCD diet with carbohydrate intake between 50 to 130 g/day is important for the long-term achievement of dietary goals.

The adverse effect on lipids is also a concern of long-term LCD. Total blood cholesterol and low-density lipoprotein (LDL) concentrations have a variable response to LCD [17]. LDL is structurally heterogenous and is further categorised based on size. DM is associated with small dense LDL (sdLDL) particles, which are associated with an increase in cardiovascular risk [18, 19]. The available evidence suggests that LCD reduces the number of sdLDL particles in obese and nondiabetic hyperlipidaemia patients [20–22]. However, no studies have investigated sdLDL among diabetic patients. In addition, atherosclerosis is regarded as the leading cause of morbidity and mortality in diabetic patients. Weight-loss diets (including LCD) can induce a nonsignificant regression of carotid intima-media thickness (IMT) in a heterogeneous group of obese patients (nondiabetes or diabetes) for two years [23]. However, no studies have focused on carotid IMT changes with moderate LCD in type 2 diabetes.

To determine the long-term effect on glycaemic control, lipids and atherosclerosis of moderate LCD on type 2 DM, we conducted an 18-month randomised clinical trial using a daily

carbohydrate intake $\leq$ 90 g/d for type 2 DM patients 20–80 years of age with either a normal body mass index (BMI) or an abnormally elevated BMI.

## Materials and methods

DOI link: http://dx.doi.org/10.17504/protocols.io.bg3zjyp6

This clinical trial was approved by the Human Research Ethics Committee of National Taiwan University Hospital (201504032RINA) on June 9, 2015. It was conducted at the Department of Family Medicine, National Taiwan University Hospital with recruitment going from February 2, 2016 to July 28, 2016 and the completion date from June 15, 2017 to January 4, 2018. All patients signed written informed consent prior to participation. This study was registered on ClinicalTrials.gov (NCT03176056); our delay in registering this study is because it is our first clinical trial. The authors confirm that all ongoing and related trials for this intervention are registered.

### Study population

Adults aged 20–80 years with type 2 DM were recruited. They were included if they had been diagnosed with diabetes for more than one year and had a poorly controlled HbA1c $\geq$ 58 mmol/mol (7.5%) in the previous three months, regardless of whether they received medications. The potential study participants were referred by physicians from outpatient clinics at the medical centre and screened by a research assistant. The exclusion criteria were pregnant or lactating women, impaired renal function with a serum creatinine $\geq$ 132.6 μmol/L (1.5 mg/dL), abnormal liver function (alanine aminotransferase (ALT), aspartate aminotransferase $\geq$3 times the normal upper limit) or liver cirrhosis, significant heart diseases (unstable angina, unstable heart failure), frequent gout attacks ($\geq$3 times/year), participation in other weight-loss programmes or the use of weight-loss drugs, eating disorders and the inability to complete the questionnaire.

### Study design

This was a single centre, parallel-designed, open-label randomised control trial, which was allocated with Taves covariate-adaptive randomisation and stratified by sex and BMI ($<$24 and $\geq$24) [24]. According to a previous study [3], the estimated absolute HbA1c reduction between the LCD and TDD groups was 0.5%, with a standard deviation (SD) of 0.408%. With a two-sided level of 5%, a power (1-ß) of 80% and an assumed 20% loss to the follow-up rate, the appropriate sample size was calculated to be 80 patients. According to a previous study [12], the estimated absolute MES reduction between the LCD and TDD groups is 0.4, with a SD of 0.5. With a two-sided level of 5%, a power (1-ß) of 80% and an assumed 20% loss to follow-up rate, the appropriate sample size was calculated to be 76 patients.

The primary outcomes were the glycaemic control status (HbA1c, fasting glucose and 2-h glucose) and the change in the medication effect score (MES). The secondary outcomes were the lipid profile, sdLDL, serum creatinine, microalbuminuria and carotid IMT.

The MES assessed the overall utilisation of antiglycaemic agents, which was computed based on the potency and dosage of diabetes medications, including insulin [10, 12]. The percentage of the maximum daily dose for each medication was multiplied by an adjustment factor, and these products were summed up to produce the final MES value. The maximum daily dose of insulin was defined as 1 unit per kilogram of baseline weight, delineating insulin resistance [12, 25]. The adjustment factors were the reported median absolute reduction in HbA1c for each medication [12] and are detailed as follows [26]: 1.5 for metformin and sulfonylureas, 2.5 for insulin, 1.0 for thiazolidinedione, 0.65 for α-glucosidase inhibitor, 0.65 for dipeptidyl

peptidase-4 inhibitors and 0.7 for sodium-glucose cotransporter 2 inhibitors [27]. For example, if a patient took 2 mg/d of glimepiride and 1500 mg/d metformin (the maximum doses for glimepiride and metformin were 8 mg/d and 3000 mg/d, respectively), the MES was calculated as 1.5×2 (mg) / 8 (mg) + 1.5 × 1500 (mg) / 3000 (mg) = 1.125, with higher values indicating a greater use of medication. The MES values were confirmed with the patients at every visit to determine their actual use.

The antidiabetic agents were categorised according to their mechanism and reported as types (categories) of diabetic medications with the number of diabetic medications (the total number of tablets and the number of insulin shots per day) at every visit. Blood pressure, weight, BMI, body composition (fat %) and waist, hip and thigh girths were measured by a research assistant every three months. Body composition was measured using a Tanita Body Composition Analyzer BC-418 (Japan). Fasting blood samples were obtained to assess fasting glucose, HbA1c, serum lipids [total cholesterol, high-densitylipoprotein (HDL), LDL, triglyceride, sdLDL], serum creatinine and 2-h blood samples for 2-h glucose levels at every visit. For the measurement of sdLDL, EDTA serum samples were assayed with the sdLDL-"Seiken" kit, a direct method for quantitative determination of sdLDL (Denka Seiken, Tokyo, Japan). The sdLDL was checked at baseline and then at six, 12, and 18 months. The microalbumin/creatinine ratio was analysed from random urine samples collected at baseline and 18 months. Complete blood cell count, uric acid, and ALT were checked at baseline and 18 months. Carotid IMT was measured at baseline and 18 months using a Toshiba SSA-550A (Japan) with a PLM-1204AT 12 MHZ probe on the longitudinal view of both common carotid arteries. The measurement was avoided for a focal elevation of more than 5 mm, a difference of elevation next to the point more than 5 mm or intimal thickness more than 15 mm [28]. Three measurements from each side were used to calculate the average carotid IMT.

## Intervention

The patients were assessed by the research physician, the research nurse and the dietitian at the first visit to record current medication use and compliance. Additionally, patients were instructed to keep a food diary for three days. Patients who complied with this requirement were evaluated and randomised by the research assistant. Individual dietary instruction was given by the research dietitian for each randomised group. Motivation group classes were arranged for both groups separately.

**Dietary intervention and surveillance.** For the LCD group, the daily carbohydrate intake was limited to less than 90 g without any restriction to the total energy. The concept of LCD with six servings of carbohydrates was introduced, and a list was provided to illustrate food items of 15 g of equivalent carbohydrates (one serving of carbohydrates). For those with good dietary compliance, sulfonylurea and insulin injections were reduced to half doses in advance to prevent hypoglycaemia.

For the TDD group, the target total calorie intake was calculated by multiplying the ideal weight by 25 kcal/kg for those with a BMI between 18.5 and 24, 20 kcal/kg for overweight/obese subjects (BMI > 24) and 30 kcal/d for underweight subjects with a BMI < 18.5. The macronutrient percentage was 50–60% for carbohydrates, 1.0–1.2 g/kg for protein and ≤ 30% for fat.

A three-day weighted food record was taken every six months. The calorie and nutrition intake of the three-day weighted food record were calculated using the E-Kitchen nutritional analysis software by a blind evaluator with enrolment numbers only.

All patients met with the research nurse every three months; these were arranged with the clinic visit, and reminders were given by phone calls.

**Physical activity surveillance.**   Exercise was recommended for both groups and was not a part of the intervention. Physical activity was assessed every three months using the International Physical Activity Questionnaire, Taiwan (IPAQ-Taiwan) [29].

**Rules for medication adjustment.**   The medication for both groups was adjusted every six months if HbA1c was higher than 64 mmol/mol (8.0%) or lower than 48 mmol/mol (6.5%), with or without hypoglycaemic symptoms.

## Statistical analysis

The analysis was performed using an intention-to-treat analysis. The participants were called back if they missed the blood test before their visits. The blood samples were reserved with another tube and provided tests if the regular samples failed. Because the participants regularly followed up at family physicians, there were no missing data. A paired t-test was conducted to compare the differences between baseline and completion of the study at 18 months within the TDD or LCD groups regarding nutrition, physical activity, glycaemic control, lipids, other blood chemistry, microalbumin/cre, carotid IMT, blood pressure, anthropometric measurements and diabetic medication. An independent t-test was used to compare the differences or 18-month mean difference (18 months minus baseline) between the TDD and LCD groups regarding the above items. The time trend of glycaemic control, MES, weight, blood pressure and lipid profile between the TDD group and the LCD group were estimated using the generalised estimating equations (GEE) method with an autoregressive (AR) covariance matrix. All analyses were conducted using SAS statistic software package 9.4 version (TS1M3 DBCS3170). A p-value $< 0.05$ was deemed statistically significant for the primary end points (HbA1c, fasting glucose, 2-h glucose and the change in the MES) in the present trial. It is noted that the analyses were applied for the secondary end points using the p-value without adjustment for multiple comparison in the present trial.

## Results

### Clinical and demographic characteristics of the enrolled participants

Ninety-two patients were recruited from February 2016 to July 2016. Eighty-five patients (92.4%) completed the study (Fig 1). There were no significant differences in the baseline characteristics between the two study groups ($p > 0.05$, Table 1).

### Changes in diet

The daily total energy intake at a particular time point showed no significant differences between the TDD and LCD groups except at six months ($p < 0.05$). The mean difference of the daily total energy intake between the baseline and particular time points (six months, 12 months and 18 months) showed a significant difference within the TDD or LCD group ($p < 0.05$).

Compared with the TDD group, the daily carbohydrate intake measured by the three-day food recall was significantly lower for the LCD group ($p < 0.05$) at every visit through 18 months. The mean difference of daily total carbohydrate intake between the baseline and particular time points (six months, 12 months and 18 months) showed a significant difference at every time period within the TDD or LCD group.

Compared with the TDD group, the daily protein and fat intake measured by the three-day food recall was significantly higher for the LCD group ($p < 0.05$) at 12 months and 18 months. Compared with the baseline, the mean difference of the daily protein intake at 12 months and 18 months was significantly higher in the LCD group ($p < 0.05$) but not in the TDD group. Compared with the baseline, the mean difference of the daily fat intake at 12 months and 18

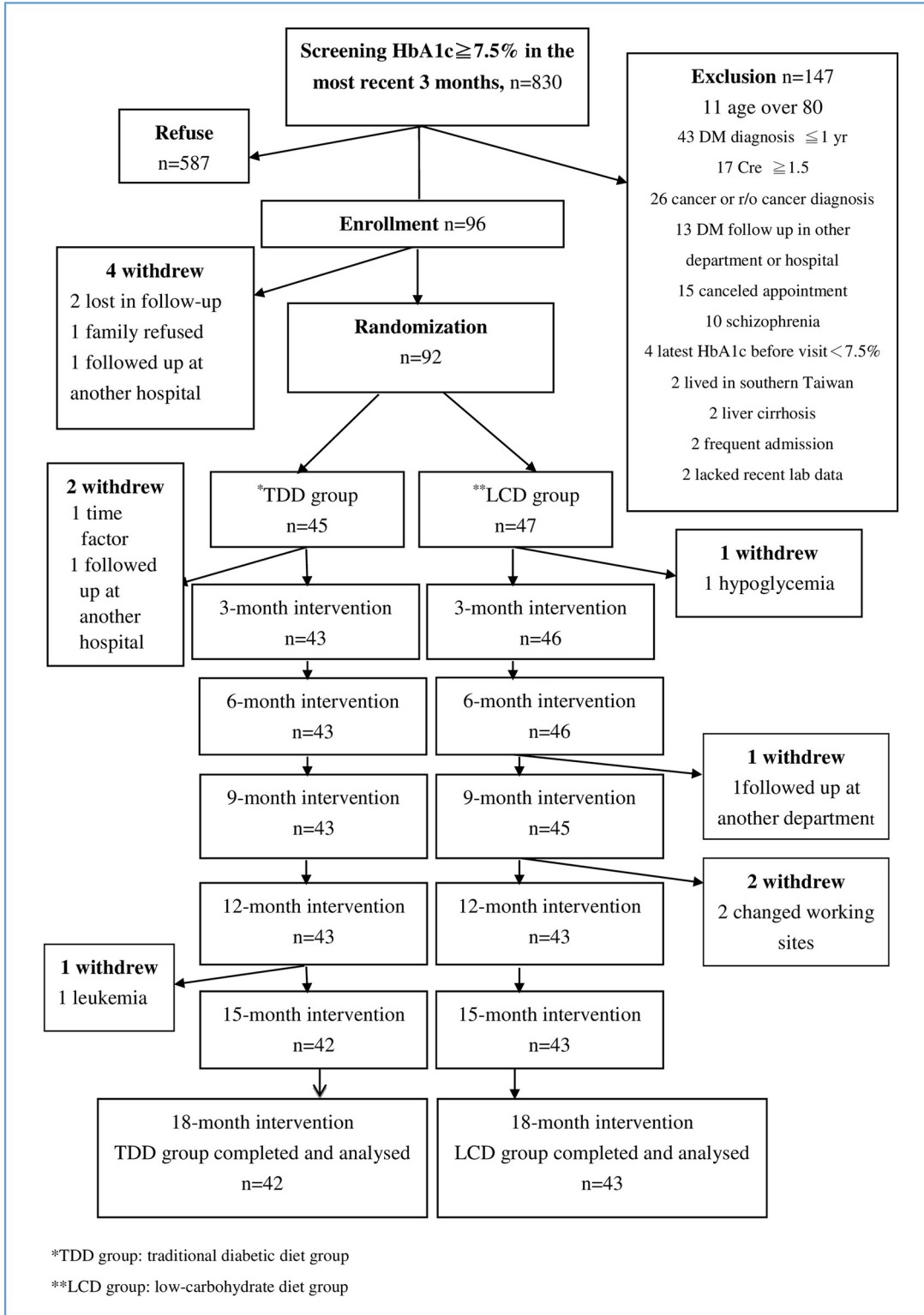

**Fig 1. Participant flow diagram.** Ninety-two patients were randomly allocated to either the TDD group or LCD group. Three patients of the TDD group and four patients of the LCD group withdrew from the study. The remaining 85 patients were followed-up for 18 months.

**Table 1. Baseline characteristics of the study participants.**

| Characteristics | TDD (n = 42) | LCD (n = 43) | p-value |
|---|---|---|---|
| | mean±SD | mean±SD | |
| Age (years) | 64.1±7.4 | 63.1±10.5 | 0.968 |
| Education (years) | 10.2±5.0 | 12.2±4.7 | 0.115 |
| Duration of diabetes (years) | 9.7±8.0 | 10.1±7.8 | 0.812 |
| | number (%) | number (%) | |
| Sex (female) | 26(61.9) | 26(60.5) | 0.706 |
| Marital status (single or widow) | 6(14.3) | 13(20.3) | 0.381 |
| Smoker | 4(9.5) | 6(11.6) | 0.952 |
| Alcohol use | 1(2.4) | 2(4.7) | 0.956 |
| BMI$\geq$24(kg/m$^2$) | 31(73.8) | 34(79.1) | 0.568 |
| Hypertension | 28(66.7) | 30(69.8) | 0.759 |
| Diabetic treatment | | | |
| Diet | 1(2.4) | 2(4.7) | |
| OHA | 31(73.8) | 37(86.0) | 0.287 |
| Insulin or (OHA with insulin) | 10(23.8) | 4(9.3) | |
| Family history of diabetes | 28(66.7) | 33(76.7) | 0.302 |

(1) TDD: traditional diabetic diet, LCD: low-carbohydrate diet

(2) *p < 0.05: significant difference between the groups by independent t-test for data of mean ± SD and by chi-square test (and Fisher's exact test if n< 5) for data of number (%).

(3) BMD: body mass index, OHA: oral hypoglycaemic agents

months was significantly higher in the LCD group (p < 0.05). However, in the TDD group, the mean difference at six months and 12 months minus baseline was significantly lower in the TDD group (p < 0.05). The LCD group consumed significantly more saturated fat (p < 0.05) and monounsaturated fat (p < 0.05) than the TDD group at most visits (Table 2) Compared with baseline, the LCD group consumed significantly more monosaturated fat at 18 months (p < 0.05), while the TDD group consumed significantly less monosaturated fat at six months (p < 0.05).

## Changes in glycaemic control

Compared with the baseline, HbA1c, fasting and 2-h glucose at 18 months were significantly decreased in the LCD group (p < 0.05). Compared with the baseline, HbA1c at 18 months was significantly decreased in the TDD group (p < 0.05), while fasting and 2-h glucose were non-significant. The mean difference of HbA1c and 2-h glucose between the baseline and 18 months was significantly greater in the LCD group compared with that of the TDD group (p < 0.05) (Table 3).

The time-group differences over 18 months showed a significant reduction for the LCD group in HbA1c, fasting glucose and 2-h glucose compared with the TDD group (p < 0.05) (HbA1c shown in Fig 2A).

## Changes in lipid profiles

Compared with the baseline, patients in the LCD group had significantly lower triglyceride and higher HDL at 18 months (p < 0.05). The 18-month changes from the baseline were not significantly different between the TDD and LCD groups regarding total cholesterol, triglyceride, LDL, HDL and sdLDL (p > 0.05) (Table 3). The time-group differences over 18 months

**Table 2. Nutritional characteristics and physical activity of the study participants.**

| Characteristics | Group | Baseline | 6m | 6m vs. Baseline MD (95% CI) | 12m | 12m vs. Baseline MD (95% CI) | 18m | 18m vs. Baseline MD (95% CI) |
|---|---|---|---|---|---|---|---|---|
| Carbohydrate (g/d) | TDD | 238.2±58.5 | 181.5±51.1* | -55.2(-70.2 ~ -40.1)† | 161.1±47.2* | -75.5(-95.2 ~ -55.9)† | 151.1±29.8* | -82.3(-100.4 ~ -64.2)† |
| | LCD | 244.1±96.1 | 123.2±54.4* | -121.7(-147.2 ~ -96.2)† | 95.5±34.6* | -148.9(-177.8 ~ -119.9)† | 88.0±29.9* | -156.9(-185.4 ~ -128.3)† |
| Protein (g/d) | TDD | 73.2±26.0 | 68.5±20.4 | -5.0(-12.9 ~ 2.9) | 67.5±16.0* | -6.0(-13.9 ~ 1.9) | 72.0±18.5* | -0.9(-9.7 ~ 8.0) |
| | LCD | 70.9±17.9 | 74.3±28.6 | 3.9(-5.0 ~ 12.8) | 82.3±23.8* | 12.0(4.7 ~ 19.3)† | 82.4.3±22.1* | 12.0(4.9 ~ 19.1)† |
| Fat (g/d) | TDD | 64.3±25.2 | 55.6±17.7 | -8.8(-16.1 ~ -1.5)† | 55.7±14.9* | -8.7(-16.3 ~ -1.1)† | 67.2±22.2* | 3.3(-7.4 ~ 14.0) |
| | LCD | 56.7±24.6 | 60.6±21.2 | 4.3(-3.8 ~ 12.4) | 69.1±19.2* | 12.3(3.9 ~ 20.8)† | 73.1±16.9* | 16.9(8.8 ~ 24.9)† |
| Polyunsaturated fat(g/d) | TDD | 9.9±7.5 | 8.1±4.4 | -1.9(-3.9 ~ 0.1) | 7.9±3.4 | -2.0(-4.0 ~ 0.0)† | 8.8±3.4 | -1.2(-3.3 ~ 0.9) |
| | LCD | 9.2±4.8 | 7.8±4.2 | -1.5(-3.2 ~ 0.2) | 9.8±4.9 | 0.4(-1.6 ~ 2.4) | 9.9±7.2 | 0.6(-1.4 ~ 2.6) |
| Monounsaturated fat (g/d) | TDD | 12.5±7.2 | 10.0±7.2* | -2.4(-4.3 ~ -0.4)† | 11.0±4.8* | -1.4(-3.4 ~ 0.6) | 12.2±4.9* | 0.3(-2.1 ~ 2.6) |
| | LCD | 12.2±7.1 | 13.0±7.9* | 0.6(-1.9 ~ 3.1) | 14.8±7.3* | 2.1(-0.5 ~ 4.7) | 17.4±7.8* | 2.9(0.4 ~ 5.5)† |
| Saturated fat(g/d) | TDD | 13.7±8.0 | 9.8±5.0 | 1.6(-1.4 ~ 4.5) | 9.8±7.1* | 1.9(-0.5 ~ 4.3) | 9.9±3.7* | 2.1(-0.3 ~ 4.4) |
| | LCD | 12.9±7.8 | 11.8±7.4 | -0.2(-2.9 ~ 2.4) | 13.2±8.3* | -0.8(-2.8 ~ 1.3) | 13.9±8.5* | 0.0(-2.0 ~ 2.1) |
| Energy (KJ/d) | TDD | 7459.7 ±1980.3 | 6176.0 ±1518.8* | -302.2(-421.9 ~ -182.5)† | 5806.6 ±1205.4 | -390.5(-520.1 ~ -260.8)† | 6145.5 ±1048.5 | -292.1(-427.6 ~ -156.6)† |
| | LCD | 7289.3 ±2153.5 | 5498.2 ±1540.5* | -422.8(-567.1 ~ -278.5)† | 5489.8 ±1212.9 | -427.6(-581.5 ~ -273.7)† | 5984.0 ±1171.1 | -350.6(-499.9 ~ -201.4)† |
| Physical activity (KJ/week) | TDD | 8582.6 ±5278.5 | 8815.3 ±6467.2 | 55.6(-299.1 ~ 410.3) | 9767.5 ±5439.2 | 283.2(-65.8 ~ 632.2) | 9899.8 ±1194.2 | 314.8(-77.0 ~ 706.7) |
| | LCD | 9617.8 ±6742.1 | 10084.7 ±6586.0 | 111.6(-244.6 ~ 467.7) | 10594.3 ±6147.6 | 233.4(-134.7 ~ 601.5) | 10599.3 ±4913.3 | 234.6(-159.9 ~ 629.1) |

(1) TDD: traditional diabetic diet, LCD: low-carbohydrate diet, MD: mean difference, M: months

(2) *p < 0.05: significant difference between the groups at specific time points by independent t-test

(3) †p < 0.05: significant difference of the mean changes at the time periods within groups by paired t-test

were not significantly different for total cholesterol, triglyceride, LDL, HDL and sdLDL between the two groups (p > 0.05) (sdLDL shown in Fig 2D).

## Changes in other laboratory profiles

Compared with the baseline, ALT at 18 months was significantly decreased in the LCD group (p < 0.05). The ALT, creatinine and uric acid showed no significant difference between the TDD and LCD groups (Table 3).

## Changes in microalbumin/cre excretion

The microalbumin/cre excretion increased for the TDD group and decreased for the LCD group over the 18-month period; however, the 18-month change from baseline in the microalbumin/cre excretion was not significantly different between the TDD and LCD groups (p > 0.05) (Table 3).

## Changes in average carotid intimal thickness

Compared with the baseline, the average carotid IMT at 18 months increased for the TDD group (p < 0.05) and remained relatively the same for the LCD group (p > 0.05). However, the 18-month mean difference from baseline in the average carotid IMT was not significantly different between the two groups (p > 0.05) (Table 3).

**Table 3. Characteristics of the participants at baseline and study completion (18 months).**

| | TDD(n = 42) | | | | | LCD(n = 43) | | | | | LCD vs. TDD (change %) |
| | baseline | | 18 months | | | baseline | | 18 months | | | |
| | mean | (SD) | mean | (SD) | [a]MD(95% CI) | mean | (SD) | mean | (SD) | [b]MD(95% CI) | p-value |
|---|---|---|---|---|---|---|---|---|---|---|---|
| Glycaemic Control | | | | | | | | | | | |
| HbA1C (%) | 8.70 | (1.01) | 7.69 | (1.06) | -1.01(-1.40 ~ -0.63)+ | 8.47 | (1.04) | 6.84 | (0.59) | -1.63(-1.96 ~ -1.30)+ | 0.0034* |
| Fasting glucose (mg/dl) | 160.17 | (37.58) | 150.48 | (37.64) | -9.69(-25.44 ~ 6.06) | 160.33 | (42.76) | 133.44 | (26.26) | -26.88(-39.47 ~ -14.29)+ | 0.0727 |
| 2-h glucose (mg/dl) | 232.76 | (65.09) | 214.07 | (72.00) | -18.69(-44.38 ~ 7.00) | 225.98 | (61.45) | 131.60 | (27.23) | -94.37(-115.23 ~ -73.51)+ | < .0001* |
| Lipids | | | | | | | | | | | |
| Total cholesterol (mg/dl) | 174.95 | (32.40) | 170.02 | (29.38) | -4.93(-15.87 ~ 6.01) | 180.12 | (34.46) | 174.88 | (31.49) | -5.23(-16.76 ~ 6.29) | 0.9737 |
| Triglyceride (mg/dl) | 177.81 | (115.77) | 160.00 | (84.50) | -17.81(-47.54 ~ 11.92) | 163.70 | (76.67) | 132.30 | (62.09) | -31.40(-55.55 ~ -7.24)+ | 0.2307 |
| LDL (mg/dl) | 103.87 | (26.17) | 100.83 | (26.09) | -3.03(-10.92 ~ 4.86) | 103.02 | (27.90) | 101.26 | (26.75) | -1.77(-10.88 ~ 7.35) | 0.5067 |
| HDL (mg/dl) | 43.61 | (9.41) | 46.55 | (8.41) | 2.94(-0.16 ~ 6.03) | 47.21 | (11.09) | 52.15 | (12.19) | 4.94(2.41 ~ 7.48)+ | 0.6157 |
| SdLDL (mg/dl) | 10.99 | (3.96) | 13.04 | (6.74) | 2.05(-0.26 ~ 4.36) | 12.69 | (6.12) | 12.72 | (5.70) | 0.03(-1.98 ~ 2.05) | 0.2696 |
| Other laboratory | | | | | | | | | | | |
| creatinine (mg/dl) | 0.89 | (0.33) | 0.97 | (0.51) | 0.08(0.00 ~ 0.17) | 0.86 | (0.23) | 0.89 | (0.22) | 0.03(-0.02 ~ 0.07) | 0.4273 |
| ALT (mg/dl) | 23.88 | (12.92) | 22.36 | (13.89) | -1.52(-5.76 ~ 2.71) | 23.51 | (15.77) | 15.56 | (5.53) | -7.95(-12.68 ~ -3.23)+ | 0.0810 |
| Uric acid(mg/dl) | 5.74 | (1.45) | 6.07 | (1.64) | 0.33(-0.07 ~ 0.73) | 5.84 | (1.42) | 6.05 | (1.14) | 0.21(-0.15 ~ 0.57) | 0.8628 |
| Microalbumin/cre (U) | 0.08 | (0.16) | 0.13 | (0.28) | 0.06(0.00 ~ 0.11) | 0.27 | (0.81) | 0.12 | (0.34) | -0.15(-0.40 ~ 0.10) | 0.1203 |
| Carotid IMT(mm) | 0.71 | (0.27) | 0.78 | (0.28) | 0.07(0.01 ~ 0.12)+ | 0.71 | (0.18) | 0.71 | (0.18) | 0.00(-0.06 ~ 0.06) | 0.0798 |
| Blood pressure | | | | | | | | | | | |
| Systolic (mmHg) | 129.83 | (11.53) | 131.45 | (11.70) | 1.62(-2.12 ~ 5.36) | 130.93 | (12.88) | 122.67 | (9.84) | -8.26(-12.94 ~ -3.58)+ | 0.0026* |
| Diastolic (mmHg) | 73.93 | (9.62) | 76.38 | (10.46) | 2.45(-0.86 ~ 5.77) | 76.79 | (9.39) | 71.79 | (7.61) | -5.00(-7.95 ~ -2.05)+ | 0.0018* |
| Anthropometric measurement | | | | | | | | | | | |
| Body weight (kg) | 68.34 | (12.29) | 67.63 | (12.48) | -0.71(-1.41 ~ -0.02)+ | 69.69 | (14.23) | 66.93 | (13.11) | -2.76(-4.64 ~ -0.88)+ | 0.0480* |
| BMI (kg/m$^2$) | 26.55 | (3.69) | 26.06 | (3.22) | -0.49(-0.87 ~ -0.11)+ | 27.31 | (4.53) | 26.11 | (4.24) | -1.20(-1.91 ~ -0.48)+ | 0.0612 |
| Fat (%) | 34.86 | (8.62) | 35.24 | (8.07) | 0.38(-0.24 ~ 0.99) | 35.77 | (7.92) | 34.60 | (8.51) | -1.17(-2.68 ~ 0.35) | 0.1379 |
| Waist circumference (cm) | 93.56 | (8.89) | 91.70 | (8.20) | -1.86(-3.34 ~ -0.37)+ | 94.06 | (11.58) | 88.37 | (8.95) | -5.69(-8.40 ~ -2.97)+ | 0.0160* |
| Hip circumference (cm) | 99.18 | (7.83) | 96.29 | (7.79) | -2.89(-4.57 ~ -1.21)+ | 100.01 | (9.91) | 93.97 | (7.53) | -6.05(-7.91 ~ -4.18)+ | 0.0143* |
| Thigh circumference (cm) | 50.36 | (7.13) | 45.64 | (4.09) | -4.72(-6.28 ~ -3.15)+ | 49.88 | (6.74) | 44.42 | (4.40) | -5.47(-7.31 ~ -3.62)+ | 0.4827 |
| Medication | | | | | | | | | | | |
| Types of diabetic medications | 2.62 | (1.08) | 2.64 | (0.93) | 0.02(-0.20 ~ 0.25) | 2.51 | (1.61) | 2.16 | (1.43) | -0.35(-0.70 ~ 0.00)+ | 0.0025* |
| Number of diabetic medications | 5.62 | (3.55) | 5.56 | (3.25) | -0.06(-0.61 ~ 0.49) | 5.12 | (4.01) | 3.99 | (3.33) | -1.13(-2.11 ~ -0.15)+ | 0.0020* |

(*Continued*)

**Table 3.** (Continued)

| | TDD(n = 42) | | | | | LCD(n = 43) | | | | | LCD vs. TDD (change %) |
|---|---|---|---|---|---|---|---|---|---|---|---|
| | baseline | | 18 months | | | baseline | | 18 months | | | |
| | mean | (SD) | mean | (SD) | [a]MD(95% CI) | mean | (SD) | mean | (SD) | [b]MD(95% CI) | p-value |
| Medication effect score (MES) | 2.13 | (1.28) | 2.08 | (1.05) | -0.05(-0.29 ~ 0.19) | 1.75 | (1.35) | 1.33 | (1.03) | -0.42(-0.74 ~ -0.09) + | 0.0018* |

(1) TDD: traditional diabetic diet, LCD: low-carbohydrate diet, MD: mean difference, M: months, LDL: low-density lipoprotein, HDL: high-density lipoprotein, sdLDL: small, dense low-density lipoprotein, ALT: alanine aminotransferase, IMT: intima-media thickness, BMI: body mass index, MES: medication effect score

(2) [a]MD: mean difference between baseline and completion of the study at 18 months in the TDD group

(3) [b]MD: mean difference between baseline and completion of the study at 18 months in the LCD group

(4) +p < 0.05: statistical significance when comparing baseline and completion of the study at 18 months using a paired t-test

(5) *p < 0.05: statistical significance when comparing [a]MD and [b]MD between TDD and LCD group using an independent t-test

## Changes in blood pressure

The 18-month mean difference from baseline showed a reduction of systolic blood pressure (SBP) and diastolic blood pressure (DBP) in the LCD group (p < 0.05) (Table 3). The mean

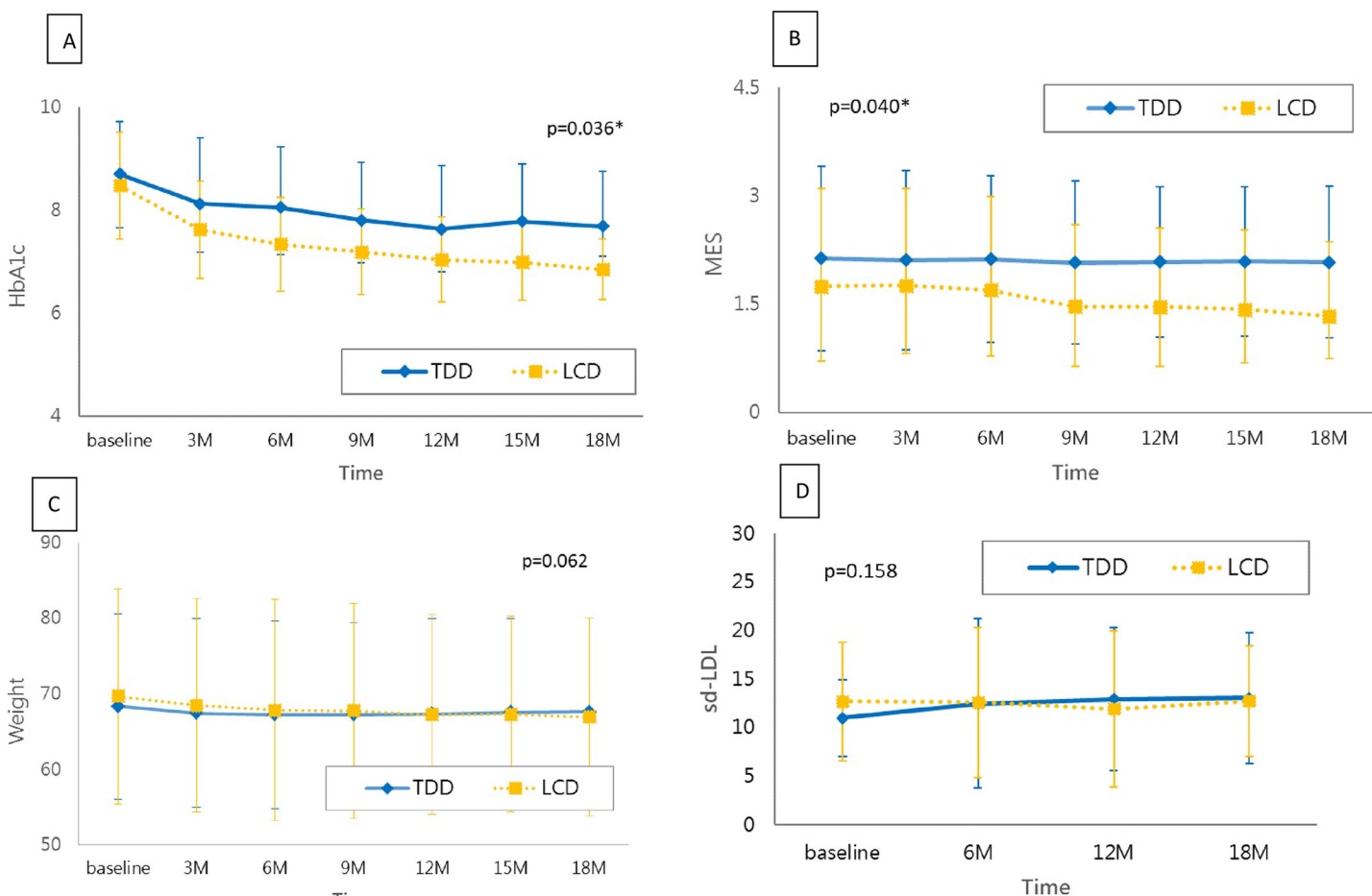

**Fig 2. 18-month changes across dietary intervention groups.** (A) HbA1c (%) and (B) MES were significantly different (p<0.05) between the TDD and LCD groups. However, the differences in (C) weight (Kg) and (D) sd-LDL (mg/dl) were not significant between the TDD and LCD groups (p>0.05).

difference of SBP and DBP between the baseline and 18 months was significantly greater in the LCD group compared with that of the TDD group ($p < 0.05$). The time-group differences over 18 months in SBP and DBP were significantly lower for LCD group ($p < 0.05$).

## Changes in anthropometric measures

Compared with the baseline, weight, BMI and waist, hip and thigh circumferences at 18 months were significantly decreased within the TDD or LCD group ($p < 0.05$). The mean difference of weight, waist circumference and hip circumference between baseline and 18 months was significantly greater in the LCD group compared with that of the TDD group ($p < 0.05$) (Table 3). The time-group differences over 18 months showed no significant difference between the LCD and the TDD group ($p > 0.05$) (Fig 2C).

## Changes in medication use

Compared with the baseline, MES and the types and number of diabetic medications at 18 months were significantly lower in the LCD group ($p < 0.05$) but not in the TDD group. Compared with the TDD group, the mean difference of MES, types and number of diabetic medications from the baseline to 18 months was significantly greater in the LCD group ($p < 0.05$) (Table 3). The time-group differences over 18 months showed a significant reduction for the LCD group in MES and number of medications compared with the TDD group ($p < 0.05$) (MES shown in Fig 2D).

## Discussion

To the best of our knowledge, our clinical trial had the highest completion rate (over 90%) when compared with similar studies [10, 11, 14, 20, 12], and it was the first clinical trial to investigate both sdLDL and carotid IMT in type 2 DM patients. The high completion rate was because these patients with chronic diseases were routinely cared for by their family physicians before enrolling in the study. They agreed with the arrangements of experimental blood tests and follow-up visits with their routine appointments without any incentives.

The intervention used ≦ six servings of carbohydrates, which was easier for the patients to understand and follow. In contrast to most LCD studies with a very low carbohydrate intake, from which patients deviated progressively over time [10, 14], the patients in the current moderate LCD study achieved their goals over time. The carbohydrate intake by the three-day weighted food record achieved the target of 90 g/d gradually at 12 and 18 months in the LCD group (Table 2). In contrast to the complicated energy calculation used for the TDD group, the patients in the LCD group only needed to consider the intake amount of carbohydrates and achieve a similar effect of calorie restriction (Table 2).

When compared with a 130 g/d LCD study [13], this 90 g/d LCD achieved twice the reduction in HbA1c, -10.8 vs. 4.8 mmol/mol (-1.2% vs. 0.65%), at six months and showed progressive effectiveness over the 18 months period. In our study, the duration of DM was 10.1±7.8 and 9.7±8.0 years for the LCD and TDD groups, respectively. Previous studies reported that poorly controlled patients with such a long history are difficult to treat; however, the present study showed that the LCD intervention resulted in a significant improvement for this group. Furthermore, we found that the LCD group had a significant reduction of the 18-month mean difference from baseline in MES, type and number of diabetic medications within group ($p < 0.05$) and between groups ($p < 0.05$). These findings indicate that the dietary effect of LCD not only reduced HbA1c but also decreased the medications required.

Although the net weight reduction was about 3 kg in the LCD group, the 18-month mean difference from the baseline was significantly reduced in weight and the waist and hip

circumference in the LCD group compared with the TDD group (p < 0.05). These findings suggest that the remodelling of fat composition existed in a small weight reduction among patients with poorly controlled type 2 DM. In addition, the small reduction in weight resulted in a significant reduction in blood pressure between the groups (p < 0.05) and normalisation of ALT in the LCD group (p < 0.05)(Table 3).

Furthermore, the LCD group had significant 18-month changes from baseline in weight and waist and hip circumference reduction (p<0.05) under daily energy intake and physical activity similar to that of the TDD group (p>0.05). This means that the LCD group was able to achieve additional weight reduction and modification of body image beyond the calorie reduction theory.

Our study did not show a reduction in sdLDL by LCD, as has been previously reported in obese and nondiabetic patients with hyperlipidaemia [17]. This might be explained by the fact that our moderate 90 g/d LCD with a higher daily carbohydrate intake than the past VLCD studies results in less effective sdLDL reduction. Conversely, the significantly greater intake of protein and fat, especially saturated fat, in the LCD group did not show adverse effects on LDL and sdLDL. Although the LCD group showed significantly lowered triglyceride and elevated HDL, these effects were not significantly different between the LCD and TDD groups. This might be explained by the moderate baseline triglyceride levels (Table 3).

The 18-month mean difference from baseline in SBP and DBP was significantly decreased for the LCD group between groups; the TDD group was increased. These findings reached statistically significant (p < 0.05) which were not observed in previous studies [10, 15], possibly because of the long-term beneficial effect.

Our study was the first to examine the change in carotid IMT after moderate LCD, and thus, it was the first to investigate the effect of moderate LCD on atherosclerosis. Although the result was not significantly different, the LCD group showed stationary carotid IMT, and the TDD group showed a slight increase over the 18-month period.

The present study is subject to several limitations. First, the number of patients with a BMI <24 was less than expected. The effectiveness of LCD on those BMI <24 patients was not analysed. Second, the opposite trends between the groups regarding microablumin excretion and the trend of higher carotid IMT increase in the TDD group indicate that a larger sample size and a longer follow-up duration may be needed. Despite these limitations, the current study has several strengths. The duration of this trial was 18 months long, with a high completion rate (> 90%), an easy-to-follow, moderate LCD guide (≦6 servings of carbohydrate), and a comprehensive list of outcomes on sdLDL, microalbuminuria, carotid IMT and MES, in addition to glycaemic control.

In conclusion, the moderate 90 g/d LCD provided better effects on glycaemic control, decreasing MES, lowering SBP/DBP, decreasing weight and reducing the waist and hip circumference than the TDD for patients with poorly controlled type 2 DM. Additionally, the moderate LCD had no adverse effects on the lipid profiles, sdLDL, serum creatinine levels, microalbumin/cre excretion, ALT or carotid IMT. Our study showed high fat, moderate protein and 90 g/d LCD resulted in better glycaemic control without adverse effects on cardiovascular risks. Hence, LCD is a reasonable dietary choice for type 2 diabetes.

## Supporting information

**S1 Checklist. Consort 20200318.**
(DOC)

**S1 File. Taiwanese LCD protocol.**
(PDF)

**S2 File. Study protocol in English.**
(PDF)

# Acknowledgments

The authors would like to thank Huang HL, Ma SM, Tsai D FC, Li CM and Chen KF from National Taiwan University Hospital for their referral of diabetic patients; the assistants Yang YC, Kuo SC and Wang RW for coordinating the trial; Chang YH, Yen YJ and Laing YC for statistical consultation from the Center of Statistical Consultation and Research in the Department of Medical Research, National Taiwan University Hospital; the volunteers Shen HY and Huang JY for group education and support; Huang CL, Chao S and volunteers from the Taiwan Health Promotion and Personnel Training Association (THPPTA) for group education. We would like to thank the International Research Promotion (www.researchpromotion.com), Department of Medical Research, National Taiwan University Hospital and Editage (www.editage.com) for English language editing.

# Author Contributions

**Conceptualization:** Chin-Ying Chen, Wei-Sheng Huang, Chin-Hao Chang.

**Data curation:** Chin-Ying Chen, Wei-Sheng Huang, Hui-Chuen Chen, Long-Teng Lee, Heng-Shuen Chen, Yow-Der Kang, Wei-Chu Chie, Chyi-Feng Jan, Wei-Dean Wang, Jaw-Shiun Tsai.

**Formal analysis:** Chin-Ying Chen, Wei-Sheng Huang, Hui-Chuen Chen, Chin-Hao Chang.

**Funding acquisition:** Chin-Ying Chen.

**Investigation:** Chin-Ying Chen, Wei-Sheng Huang, Hui-Chuen Chen.

**Methodology:** Chin-Ying Chen, Wei-Sheng Huang, Hui-Chuen Chen, Chin-Hao Chang.

**Project administration:** Chin-Ying Chen.

**Resources:** Chin-Ying Chen, Wei-Sheng Huang, Hui-Chuen Chen.

**Software:** Chin-Ying Chen, Chin-Hao Chang.

**Supervision:** Chin-Ying Chen, Wei-Sheng Huang, Chin-Hao Chang.

**Validation:** Chin-Ying Chen, Wei-Sheng Huang, Chin-Hao Chang.

**Visualization:** Chin-Ying Chen, Wei-Sheng Huang, Chin-Hao Chang.

**Writing – original draft:** Chin-Ying Chen.

**Writing – review & editing:** Chin-Ying Chen.

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
