## [Decision Letter · Decision Letter 0]

1 Jun 2020

PONE-D-20-08376

Effect of a 90 g/day low-carbohydrate diet on glycemic control, small dense low density lipoprotein, and carotid intima-media thickness in type 2 diabetic patients: an 18month randomized controlled trial

PLOS ONE

Dear Dr. Wei Sheng Huang,

Thank you for submitting your manuscript to PLOS ONE. After careful consideration, we feel that it has merit but does not fully meet PLOS ONE’s publication criteria as it currently stands. Therefore, we invite you to submit a revised version of the manuscript that addresses the points raised during the review process.

We look forward to receiving your revised manuscript.

Kind regards,

Elena Barengolts, MD

Academic Editor

PLOS ONE

2. Thank you for submitting your clinical trial to PLOS ONE and for providing the name of the registry and the registration number. The information in the registry entry suggests that your trial was registered after patient recruitment began. PLOS ONE strongly encourages authors to register all trials before recruiting the first participant in a study.

1) your reasons for your delay in registering this study (after enrolment of participants started);

2) confirmation that all related trials are registered by stating: “The authors confirm that all ongoing and related trials for this drug/intervention are registered”.

Please also ensure you report the date at which the ethics committee approved the study as well as the complete date range for patient recruitment and follow-up in the Methods section of your manuscript."

3. Please include a caption for figures 1 and 2.

Additional Editor Comments:

1.Paper is substantially improved but additional improvements are needed. The paper needs improvement in English language. Some examples are below, however, there are some others.

2.Attention to details: for example, use of designations under the Tables “p-value^c^” is usually done as “^c^p-value”, etc. Look at some other published paper and do appropriate changes.

3.Was this blinded trial? Did patients, researchers, and statisticians new group assignment? Add this part to the Methods. If trial was not blinded, clearly state so in Methods.

4.Was the trial registered in National registry? If not, please state so in the Methods.

5.Table 2: “p-value: the difference between group at specific time point, *p < 0.05”. Do you mean “between groups”? Need to add which statistical method was used, and whether adjustment for multiple comparisons was made.

6.Table 2: line 263 “p-value: the difference…” has to be *p-value…

7.Table 3: The name of the Table and statistics are not clear. For example, can say: “Characteristics of participants at baseline and study completion (18 months)”.

8.Table 3: “p-value^a^: : The 18-month change”. Not clear: Is this comparison between Baseline and 18-mo absolute values? Is so, then change explanation under the Table. Correct also for “p-value^b^. It probably should say “^a^p-value: comparison within the group between Baseline and Completion of the study at 18-mo, using…test” or similar explanation.

9.Table 3, line 260: explain what “change” means, such as “change (18-mo minus baseline)” or similar explanation.

10.Fig. 2: How did you chose characteristics? A,B,D choice is appropriate (primary and secondary outcomes). Why DBP? It’s more appropriate show microalbuminuria, and/or carotid intima-media thickness since these are important secondary outcomes or weight since this is an important clinical outcome.

11.2/34: English not correct, use “respectively” instead of ‘separately’.

12.5/91: English needs improvement: “…patients (including diabetes) for 2 years”

13.5/165: Need clarification (English not correct): “…evaluated by the first assessment” Do you mean: evaluated by the same research team members?

14.5/165-167: Need clarification (English not correct): “Patients who complied with this requirement were evaluated by the first assessment and randomized by the research assistant after the first assessment.”

15.10/210: Table 1: Each characteristic needs units of measurements, for example, “Age, years”, Sex, n (%), etc.

16.6/171: “ 6 units of carbohydrate”. Do you mean “servings”? If so, change at all places.

Ensure all the important points raised by the reviewers are incorporated into the manuscript.

Reviewers' comments:

Reviewer's Responses to Questions

**Comments to the Author**

1. Is the manuscript technically sound, and do the data support the conclusions?

Reviewer #1: Partly

Reviewer #2: Yes

2. Has the statistical analysis been performed appropriately and rigorously? 

Reviewer #1: No

Reviewer #2: Yes

3. Have the authors made all data underlying the findings in their manuscript fully available?

Reviewer #1: Yes

Reviewer #2: Yes

4. Is the manuscript presented in an intelligible fashion and written in standard English?

Reviewer #1: Yes

Reviewer #2: Yes

5. Review Comments to the Author

Reviewer #1: The manuscript entitled ‘Effect of a 90 g/day low-carbohydrate diet on glycemic control, small dense low density lipoprotein, and carotid intima-media thickness in type 2 diabetic patients: an 18 month randomized controlled trial’ with the aim to determine the effect of a moderate (90 g/d) low carbohydrate diet (LCD) in type 2 diabetes patients over 18 months.

The manuscript can be further improved based on the following comments.

Abstracts

For Results, the word mean sd to be stated where applicable.

Materials and Methods

Page 6 Line 120 allocation concealment information to be stated.

Sample size calculation

Page 6 Line 125-126, there were 4 primary outcomes namely glycemic control status (HbA1c, fasting glucose, and 2-h glucose) and the change in the medication effect score (MES). Was the sample size calculation took consideration of the other primary outcomes such as MES?

Statistical Analysis

Page 9 Line 197-199, word mean to be added to describe the use of independent t test and paired t test.

More information on the missing data i.e percentage/type of missing data to be provided.

Page 9 Line 200- 201, the use of GEE for what comparison, time points, it's assumptions, working correlation structure etc to be clearly stated in the statistical analysis section. The results of GEE analysis to be clearly highlighted in the results section including detail results, goodness of fit etc.

Page 9 Line 202, proper citation for SAS to be provided.

Results

Page 10 Table 1, the statistical tests which were used in the analysis to be denoted in the table footnote. Nonetheless, based on CONSORT requirements all baseline comparison to be avoided.

Page 12 Table 2. the focus of the analysis to be more on within group comparison rather than comparison between groups at a particular time point. The mean changes between the time period can be compared between the groups i.e. mean difference (baseline to 6 months), mean difference (baseline to 12 months) and mean difference (baseline to 18 months). 95% confidence interval to be provided apart from p value. Statistical test to be denoted in the table footnote.

Page 11 Line 212 to 218, the description of the results to be revised accordingly.

Page 13 Line 240-247, the paragraphs not clear and confusing. In Line 241-243, it was stated HbA1c deceased in both TDD and LCD group at 18 months (but at two separate sentence). There were two HbA1c (mmol/mol and %) in Table 3 and need to be clearly stated in the paragraph, .

Page 14 Table 3, technically p value cannot be zero (to use symbol < ). Height data to be stated. Statistical test to be denoted in the table footnote. Mean difference (pre-post)/effect size for within group as well as 95% confidence interval to be provided.

Page 16 Line 277-278, the sentence requires improvement. Creatinine and uric acid was not statistically significant (baseline to 18 months) for LCD group but mean differences (baseline-18 months) between the groups was statistically significant for creatinine.

Page 17 Line 299-301, the sentences are confusing especially involving description of weight, BMI and thigh circumference.

Page 18 Line 310, p value to be stated.

Page 23 Reference No. 16, et al to be used for more than 6 authors.

Ensure all the important points raised by the reviewers are incorporated into the manuscript.

Reviewer #2: I think the manuscript is much improved. Very interesting dataset and will make a valuable addition to the evidence-ase. Only a couple of grammar points from me:

Line 214: daily carbohydrate intake 'was' instead of 'were'

Line 217'consumed' instead of 'took' fat

6. PLOS authors have the option to publish the peer review history of their article (what does this mean?). If published, this will include your full peer review and any attached files.

Reviewer #1: No

Reviewer #2: No

---

## [Author Response · Author response to Decision Letter 0]

15 Jul 2020

Academic Editor

1. Answer: Thank you for your reminder. There are no changes of our financial disclosure.

Answer: Thank you for your reminder. We add it in line 99.

Academic Editor

PLOS ONE Journal requirements:

Answer: Thank you for your reminder. We do.

2. Thank you for submitting your clinical trial to PLOS ONE and for providing the name of the registry and the registration number. The information in the registry entry suggests that your trial was registered after patient recruitment began. PLOS ONE strongly encourages authors to register all trials before recruiting the first participant in a study.

1) your reasons for your delay in registering this study;

Answer: Thank you very much for your comments. Our delay in registering this study is because it is our first clinical trial. (line 105-6)

2) confirmation that all related trials are registered by stating: “The authors confirm that all ongoing and related trials for this drug/intervention are registered”. 

Answer: Thank you very much for your comments. We add it in line 106-7.

Please also ensure you report the date at which the ethics committee approved the study as well as the complete date range for patient recruitment and follow-up in the Methods section of your manuscript." 

Answer: Thank you very much for your comments. This clinical trial was approved by the Human Research Ethics Committee of National Taiwan University Hospital (201504032RINA) on June 9, 2015. It was conducted at the Department of Family Medicine, National Taiwan University Hospital with recruitment going from February 2, 2016 to July 28, 2016 and the completion date from June 15, 2017 to January 4, 2018. (line 100-4) .

3. Please include a caption for figures 1 and 2. 

Answer: Thank you very much for your comments. Figure 1: Participant flow diagram. Figure 2: 18-month changes across dietary intervention groups.

Additional Editor Comments:

1. Paper is substantially improved but additional improvements are needed. The paper needs improvement in English language. Some examples are below, however, there are some others. 

Answer: Thank you very much for your comments. We have sent for English editing again. International Research Promotion (www.researchpromotion.com) (line417-8)

2. Attention to details: for example, use of designations under the Tables “p-valuec” is usually done as “cp-value”, etc. Look at some other published paper and do appropriate changes. 

Answer: Thank you very much for your comments. The designations were changed in Table 3.

3. Was this blinded trial? Did patients, researchers, and statisticians new group assignment? Add this part to the Methods. If trial was not blinded, clearly state so in Methods. 

Answer: Thank you very much for your comments. This was a single center, parallel-designed, open-label randomized control trial.(line 123)

4. Was the trial registered in National registry? If not, please state so in the Methods.

Answer: Thank you very much for your comments. This study was registered on ClinicalTrials.gov (NCT03176056). (line 104)

5.Table 2: “p-value: the difference between group at specific time point, *p < 0.05”. Do you mean “between groups”? Need to add which statistical method was used, and whether adjustment for multiple comparisons was made. 

Answer: Thank you very much for your comments. (line 259-260)

(2) *p < 0.05: significant difference between the groups at specific time points by independent t-test

(3) †p < 0.05: significant difference of the mean changes at the time periods within groups by paired t-test

6.Table 2: line 263 “p-value: the difference…” has to be *p-value…

Answer: Thank you very much for your comments. The word was omitted

7.Table 3: The name of the Table and statistics are not clear. For example, can say: “Characteristics of participants at baseline and study completion (18 months)”.

Answer: Thank you very much for your comments. Table 3. Characteristics of participants at baseline and study completion (18 months) (line 278)

8.Table 3: “p-valuea: : The 18-month change”. Not clear: Is this comparison between Baseline and 18-mo absolute values? Is so, then change explanation under the Table. Correct also for “p-valueb. It probably should say “ap-value: comparison within the group between Baseline and Completion of the study at 18-mo, using…test” or similar explanation.

Answer: Thank you very much for your comments. They are corrected. 

(4) +p < 0.05: statistical significance when comparing baseline and completion of the study at 18 months using a paired t-test

(5) *p < 0.05: statistical significance when comparing aMD and bMD between TDD and LCD group using an independent t-test

( line 278-286)

9.Table 3, line 260: explain what “change” means, such as “change (18-mo minus baseline)” or similar explanation.

Answer: Thank you very much for your comments. It is corrected

(2) aMD: mean difference between baseline and completion of the study at 18 months in the TDD group 

(3)bMD: mean difference between baseline and completion of the study at 18 months in the LCD grou . (line 278-286)

10.Fig. 2: How did you chose characteristics? A,B,D choice is appropriate (primary and secondary outcomes). Why DBP? It’s more appropriate show microalbuminuria, and/or carotid intima-media thickness since these are important secondary outcomes or weight since this is an important clinical outcome.

Answer: Thank you very much for your comments. It is difficult to choose from many characteristics from different spectrums. Microalbuminuria and carotid intima-media thickness are not selected because these two characteristics are measured twice (baseline and 18 months) and easy to show the result by simple tables. We selected those characteristics with 7 measurements ( baseline and every 3 months, up to 18 months) or 4 measurements ( baseline and every 6 months, up to 18 months). The DBP was selected because of the effectiveness of DBP reduction in our trial. We follow your recommendation and substitute weight loss for DBP because weight is an important outcome.

11.2/34: English not correct, use “respectively” instead of ‘separately’.

Answer: Thank you very much for your comments. The LCD and the TDD group consumed 88.0±29.9 g and 151.1±29.8 g of carbohydrate respectively. (line 34)

12.5/91: English needs improvement: “…patients (including diabetes) for 2 years”

Answer: Thank you very much for your comments. We correct it as “obesity patient (non-diabetes or diabetes)”.(line 91)

13.5/165: Need clarification (English not correct): “…evaluated by the first assessment” Do you mean: evaluated by the same research team members?

Answer: Thank you very much for your comments. Patients who complied with this requirement were evaluated and randomized by the research assistant.(line 171-2)

14.5/165-167: Need clarification (English not correct): “Patients who complied with this requirement were evaluated by the first assessment and randomized by the research assistant after the first assessment.”

Answer: Thank you very much for your comments. Patients who complied with this requirement were evaluated and randomized by the research assistant.(line 171-2)

15.10/210: Table 1: Each characteristic needs units of measurements, for example, “Age, years”, Sex, n (%), etc.

Answer: Thank you very much for your comments.Age, years; Duration of diabetes, years; others: mention in note (2)Data are mean ± SD or number (%).(line 230)

16.6/171: “ 6 units of carbohydrate”. Do you mean “servings”? If so, change at all places.

Answer: Thank you very much for your comments. Units were changed to servings. (line 178, 180,345,398)

17.Ensure all the important points raised by the reviewers are incorporated into the manuscript.

Reviewers' comments:

Reviewer's Responses to Questions

Comments to the Author

1. Is the manuscript technically sound, and do the data support the conclusions?

Reviewer #1: Partly

Reviewer #2: Yes

2. Has the statistical analysis been performed appropriately and rigorously? 

Reviewer #1: No

Reviewer #2: Yes

3. Have the authors made all data underlying the findings in their manuscript fully available?

Reviewer #1: Yes

Reviewer #2: Yes

4. Is the manuscript presented in an intelligible fashion and written in standard English?

Reviewer #1: Yes

Reviewer #2: Yes

5. Review Comments to the Author

Reviewer #1: The manuscript entitled ‘Effect of a 90 g/day low-carbohydrate diet on glycemic control, small dense low density lipoprotein, and carotid intima-media thickness in type 2 diabetic patients: an 18 month randomized controlled trial’ with the aim to determine the effect of a moderate (90 g/d) low carbohydrate diet (LCD) in type 2 diabetes patients over 18 months.

The manuscript can be further improved based on the following comments.

Abstracts

For Results, the word mean sd to be stated where applicable.

Answer: Thank you very much for your comments. The 18-month mean change from baseline was statistically significant for the HbA1c (-1.6±0.3 vs. -1.0±0.3%), 2-h glucose (-94.4±20.8 vs. -18.7±25.7 mg/dl), MES (-0.42±0.32 vs. -0.05±0.24), weight (-2.8±1.8 vs. -0.7±0.7 kg), waist circumference (-5.7±2.7 vs. -1.9±1.4 cm), hip circumference (-6.1±1.8 vs. -2.9±1.7 cm) and blood pressure (-8.3±4.6/-5.0±3 vs. 1.6±0.5/2.5±1.6 mmHg) between the LCD and TDD groups (p<0.05). (Line35-39)

Materials and Methods

Page 6 Line 120 allocation concealment information to be stated.

Answer: This was a single center, parallel-designed, open-label randomized control trial.(line 123)

Sample size calculation

Page 6 Line 125-126, there were 4 primary outcomes namely glycemic control status (HbA1c, fasting glucose, and 2-h glucose) and the change in the medication effect score (MES). Was the sample size calculation took consideration of the other primary outcomes such as MES?

Answer: Thank you very much for your comments. According to a previous study [12], the estimated absolute MES reduction between the LCD and TDD groups is 0.4, with a SD of 0.5. With a two-sided level of 5%, a power (1-ß) of 80% and an assumed 20% loss to follow-up rate, the appropriate sample size was calculated to be 76 patients .(line 128-31)

Statistical Analysis

Page 9 Line 197-199, word mean to be added to describe the use of independent t test and paired t test.

Answer: Thank you very much for your comments. A paired t-test was conducted to compare the differences between baseline and completion of the study at 18 months within the TDD or LCD groups regarding nutrition, physical activity, glycaemic control, lipids, other blood chemistry, microalbumin/cre, carotid IMT, blood pressure, anthropometric measurements and diabetic medication. An independent t-test was used to compare the differences or 18-month mean difference (18 months minus baseline) between the TDD and LCD groups regarding the above items. (line 207-213)

More information on the missing data i.e percentage/type of missing data to be provided.

Answer: Thank you very much for your comments. The participants were called back if they missed the blood test before their visits. The blood samples were reserved with another tube and provided tests if the regular samples failed. Because the participants regularly followed up at family physicians, there were no missing data. (line 205-8)

Page 9 Line 200- 201, the use of GEE for what comparison, time points, it's assumptions, working correlation structure etc to be clearly stated in the statistical analysis section. The results of GEE analysis to be clearly highlighted in the results section including detail results, goodness of fit etc.

Answer: Thank you very much for your comments. The time trend of glycaemic control, MES, weight, blood pressure and lipid profile between the TDD group and the LCD group were estimated using the generalised estimating equations (GEE) method with an autoregressive (AR) covariance matrix. (line 213-6)

Page 9 Line 202, proper citation for SAS to be provided.

Answer: Thank you very much for your comments. SAS 9.4 version (TS1M3 DBCS3170).(line 216-7)

Results

Page 10 Table 1, the statistical tests which were used in the analysis to be denoted in the table footnote. Nonetheless, based on CONSORT requirements all baseline comparison to be avoided.

Answer: Thank you very much for your comments. Data are mean ± SD or number (%); independent t-test or chi-square test (line 230)

Page 12 Table 2. the focus of the analysis to be more on within group comparison rather than comparison between groups at a particular time point. The mean changes between the time period can be compared between the groups i.e. mean difference (baseline to 6 months), mean difference (baseline to 12 months) and mean difference (baseline to 18 months). 95% confidence interval to be provided apart from p value. Statistical test to be denoted in the table footnote.

Answer: Thank you very much for your comments. The table 2 was updated.(line 238). The statistical test was denoted in table footnote. (2) *p < 0.05: significant difference between the groups at specific time points by independent t-test

(3)†p < 0.05: significant difference of the mean changes at the time periods within groups by paired t-test (line 258-260)

Page 11 Line 212 to 218, the description of the results to be revised accordingly.

Answer: Thank you very much for your comments. It is revised in line 233-255.

The daily total energy intake at a particular time point showed no significant differences between the TDD and LCD groups except at six months (p < 0.05). The mean difference of the daily total energy intake between the baseline and particular time points (six months, 12 months and 18 months) showed a significant difference within the TDD or LCD group (p < 0.05).

Compared with the TDD group, the daily carbohydrate intake measured by the three-day food recall was significantly lower for the LCD group (p < 0.05) at every visit through 18 months. The mean difference of daily total carbohydrate intake between the baseline and particular time points (six months, 12 months and 18 months) showed a significant difference at every time period within the TDD or LCD group. 

Compared with the TDD group, the daily protein and fat intake measured by the three-day food recall was significantly higher for the LCD group (p < 0.05) at 12 months and 18 months. Compared with the baseline, the mean difference of the daily protein intake at 12 months and 18 months was significantly higher in the LCD group (p < 0.05) but not in the TDD group. Compared with the baseline, the mean difference of the daily fat intake at 12 months and 18 months was significantly higher in the LCD group (p < 0.05). However, in the TDD group, the mean difference at six months and 12 months minus baseline was significantly lower in the TDD group (p < 0.05) . The LCD group consumed significantly more saturated fat (p < 0.05) and monounsaturated fat (p < 0.05) than the TDD group at most visits (Table 2) Compared with baseline, the LCD group consumed significantly more monosaturated fat at 18 months (p < 0.05), while the TDD group consumed significantly less monosaturated fat at six months (p < 0.05)

Page 13 Line 240-247, the paragraphs not clear and confusing. In Line 241-243, it was stated HbA1c deceased in both TDD and LCD group at 18 months (but at two separate sentence). There were two HbA1c (mmol/mol and %) in Table 3 and need to be clearly stated in the paragraph, .

Answer: Thank you very much for your comments. It was reported in two sentence because 3 items (HbA1c, fasting and 2-h glucose) were significant for LCD group, but 1 item ( HbA1c) was significant for TDD group. (line 262-267).The duplicated Hb A1c was omitted (table 3).

Page 14 Table 3, technically p value cannot be zero (to use symbol < ). Height data to be stated. Statistical test to be denoted in the table footnote. Mean difference (pre-post)/effect size for within group as well as 95% confidence interval to be provided.

Answer: Thank you very much for your comments. The table 3 is redone. Height data was not added because it was not changed during the 18-month period. Statistical test, mean difference (pre-post)/effect size and 95% confidence interval were provided in table 3.(line 277)

Page 16 Line 277-278, the sentence requires improvement. Creatinine and uric acid was not statistically significant (baseline to 18 months) for LCD group but mean differences (baseline-18 months) between the groups was statistically significant for creatinine.

Answer: Thank you very much for your comments. The p-value between groups for creatinine was mistyped. It is non-significant between the groups.(line 277)

Page 17 Line 299-301, the sentences are confusing especially involving description of weight, BMI and thigh circumference.

Answer: Thank you very much for your comments. It is revised. 

Compared with the baseline, weight, BMI and waist, hip and thigh circumferences at 18 months were significantly decreased within the TDD or LCD group (p < 0.05). The mean difference of weight, waist circumference and hip circumference between baseline and 18 months was significantly greater in the LCD group compared with that of the TDD group (p < 0.05) (Table 3). The time-group differences over 18 months showed no significant difference between the LCD and the TDD group (p > 0.05) (Figure 2-C). (line 323-328)

Page 18 Line 310, p value to be stated.

Answer: Thank you very much for your comments. Compared with the TDD group, the mean difference of MES, types and number of diabetic medications from the baseline to 18 months was significantly greater in the LCD group (p < 0.05) (Table 3).(line 333-335)

Page 23 Reference No. 16, et al to be used for more than 6 authors.

Answer: Thank you very much for your comments. 16. It is.

Guldbrand H, Dizdar B, Bunjaku B, et al: In type 2 diabetes, randomisation to advice to follow a low-carbohydrate diet transiently improves glycaemic control compared with advice to follow a low-fat diet producing a similar weight loss. Diabetologia 2012; 55:2118-27. (line 474)

Ensure all the important points raised by the reviewers are incorporated into the manuscript.

Reviewer #2: I think the manuscript is much improved. Very interesting dataset and will make a valuable addition to the evidence-ase. Only a couple of grammar points from me:

Line 214: daily carbohydrate intake 'was' instead of 'were'

Answer: Thank you very much for your comments. The daily carbohydrate intake measured by the 3-day food recall was significant lower for the LCD group (both p < 0.001) at every visits through the 18 months.(line 240)

Line 217'consumed' instead of 'took' fat

Answer: Thank you very much for your comments. The LCD group consumed significant more saturated fat (p < 0.05) and monounsaturated fat (p < 0.05) than the TDD group at most visits(line 251)

6. PLOS authors have the option to publish the peer review history of their article (what does this mean?). If published, this will include your full peer review and any attached files.

Do you want your identity to be public for this peer review? For information about this choice, including consent withdrawal, please see our Privacy Policy.

Reviewer #1: No

Reviewer #2: No

---

## [Decision Letter · Decision Letter 1]

30 Jul 2020

PONE-D-20-08376R1

Effect of a 90 g/day low-carbohydrate diet on glycaemic control, small, dense low-density lipoprotein and carotid intima-media thickness in type 2 diabetic patients: an 18-month randomised controlled trial

PLOS ONE

Dear Dr. Huang,

Thank you for submitting your manuscript to PLOS ONE. After careful consideration, we feel that it has merit but does not fully meet PLOS ONE’s publication criteria as it currently stands. Therefore, we invite you to submit a revised version of the manuscript that addresses the points raised during the review process.

Please respond to the minor comments from Reviewer #1.

We look forward to receiving your revised manuscript.

Kind regards,

Elena Barengolts, MD

Academic Editor

PLOS ONE

Reviewers' comments:

Reviewer's Responses to Questions

**Comments to the Author**

1. If the authors have adequately addressed your comments raised in a previous round of review and you feel that this manuscript is now acceptable for publication, you may indicate that here to bypass the “Comments to the Author” section, enter your conflict of interest statement in the “Confidential to Editor” section, and submit your "Accept" recommendation.

Reviewer #1: (No Response)

2. Is the manuscript technically sound, and do the data support the conclusions?

Reviewer #1: Yes

3. Has the statistical analysis been performed appropriately and rigorously? 

Reviewer #1: (No Response)

4. Have the authors made all data underlying the findings in their manuscript fully available?

Reviewer #1: Yes

5. Is the manuscript presented in an intelligible fashion and written in standard English?

Reviewer #1: Yes

6. Review Comments to the Author

Reviewer #1: Minor comments

Table 1, it is best to provide symbol n (%) in the table. It may difficult for readers to identify which n(%) and mean (sd). Likewise the statistical tests to be denoted with symbol i.e. *, + etc

Table 2, for the multiple comparison, were adjustment made to the p value? If not, the reason to be clearly stated in the statistical analysis section.

7. PLOS authors have the option to publish the peer review history of their article (what does this mean?). If published, this will include your full peer review and any attached files.

Reviewer #1: No

---

## [Author Response · Author response to Decision Letter 1]

2 Sep 2020

Reviewer #1:?Minor comments

Table 1, it is best to provide symbol n (%) in the table. It may difficult for readers to identify which n(%) and mean (sd). Likewise the statistical tests to be denoted with symbol i.e. *, + etc

Answer: Thank you very much for your comments. N(%) and mean(SD) were provided in the table 1 . The statistical tests were denoted with symbol *. 

(2) *p < 0.05: significant difference between the groups by independent t-test for data of mean ± SD and by chi-square test (and Fisher’s exact test if n< 5) for data of number (%) (line 233).

Table 2, for the multiple comparison, were adjustment made to the p value? If not, the reason to be clearly stated in the statistical analysis section.

Answer: Thank you very much for your comments.

A p-value < 0.05 was deemed statistically significant for the primary end points (HbA1c, fasting glucose, 2-h glucose and the change in the MES) in the present trial. It is noted that the analyses were applied for the secondary end points using the p-value without adjustment for multiple comparison in the present trial.(line 218-221)

---

## [Decision Letter · Decision Letter 2]

22 Sep 2020

Effect of a 90 g/day low-carbohydrate diet on glycaemic control, small, dense low-density lipoprotein and carotid intima-media thickness in type 2 diabetic patients: an 18-month randomised controlled trial

PONE-D-20-08376R2

Dear Dr. Huang,

We’re pleased to inform you that your manuscript has been judged scientifically suitable for publication and will be formally accepted for publication once it meets all outstanding technical requirements.

Kind regards,

Elena Barengolts, MD

Academic Editor

PLOS ONE

Additional Editor Comments (optional):

Reviewers' comments:

Reviewer's Responses to Questions

**Comments to the Author**

1. If the authors have adequately addressed your comments raised in a previous round of review and you feel that this manuscript is now acceptable for publication, you may indicate that here to bypass the “Comments to the Author” section, enter your conflict of interest statement in the “Confidential to Editor” section, and submit your "Accept" recommendation.

Reviewer #1: (No Response)

2. Is the manuscript technically sound, and do the data support the conclusions?

Reviewer #1: Yes

3. Has the statistical analysis been performed appropriately and rigorously? 

Reviewer #1: (No Response)

4. Have the authors made all data underlying the findings in their manuscript fully available?

Reviewer #1: Yes

5. Is the manuscript presented in an intelligible fashion and written in standard English?

Reviewer #1: Yes

6. Review Comments to the Author

Reviewer #1: (No Response)

7. PLOS authors have the option to publish the peer review history of their article (what does this mean?). If published, this will include your full peer review and any attached files.

Reviewer #1: No

---

## [Editor Report · Acceptance letter]

25 Sep 2020

PONE-D-20-08376R2 

Effect of a 90 g/day low-carbohydrate diet on glycaemic control, small, dense low-density lipoprotein and carotid intima-media thickness in type 2 diabetic patients: an 18-month randomised controlled trial 

Dear Dr. Huang:

I'm pleased to inform you that your manuscript has been deemed suitable for publication in PLOS ONE. Congratulations! Your manuscript is now with our production department. 

Kind regards, 

on behalf of

Dr. Elena Barengolts 

Academic Editor

PLOS ONE